# The Effectiveness of Selected Devices to Reduce the Speed of Vehicles on Pedestrian Crossings

**Maciej Kruszyna** [1,*] and **Marta Matczuk-Pisarek** [2]

1 Faculty of Civil Engineering, Wrocław University of Science and Technology, 50-370 Wrocław, Poland
2 Urząd Miasta Jelenia Góra, Referat Komunikacji Miejskiej i Zarządzania Ruchem, 58-500 Jelenia Góra, Poland; mmatczukpisarek@gmail.com
* Correspondence: maciej.kruszyna@pwr.edu.pl

**Abstract:** Accidents involving pedestrians often result in serious injury or death. The main goal of this conducted research is to evaluate selected devices that will help reduce the speed of vehicles on pedestrian crossings. Many devices from a group of "speed control measures" and "mid block tools" (refugee islands, speed tables, and raised pedestrian crossings) are examined to find the most effective ones. In our research, the range of reduction of a vehicle's speed is used as a main measure of effectiveness, but a wider statistical analysis was conducted as well. One of the results of the research is the identification of three categories of devices referred to as high effectives (good), medium effectives (intermediate), and low or lack of effectives (bad). The content of the paper starts by highlighting the reasons to reduce the vehicle's speed on pedestrian crossings (as an introduction). Next, we present the description of devices used to reduce the vehicle's speed with a presentation of the research of their effectiveness. The studies that have been conducted are described in the following chapters: first, the characteristic of method and location, second, with discussion, the results of research and identification of the three categories of devices. The paper is then summarized by conclusions and comments. The research only covered the issues of road traffic engineering. The research was made in Poland, but the conclusions could be useful worldwide due to similar traffic rules and technical solutions.

**Keywords:** speed control devices; pedestrian crossings; pedestrians safety; traffic measurements; speed distribution

## 1. Introduction—Reasons to Reduce the Vehicles Speed on Pedestrian Crossings

The main goal of the conducted research is an evaluation of selected devices in the aspect of reducing vehicles' speed on pedestrian crossings. Highlighting the reasons to reduce the vehicles speed on pedestrian crossings as a background to the next parts of the research is done first.

Accidents involving pedestrians often result in serious injury or death to these "less protected road users". There is an increase in the number of accidents involving pedestrians [1]. About 1/3 of fatalities are pedestrians [2]. About $\frac{1}{4}$ of pedestrian deaths in Europe occur on crossings [3]. In India, 22% of road deaths are related to pedestrians [4], and in North Carolina, 10–20% of pedestrian accidents are fatal [5]. In the Dhaka metropolitan area (Bangladesh), about 65% of fatal collisions are with pedestrians [6]. In Louisiana, there was a greater increase in the number of accidents involving pedestrians than the increase in the total number of accidents [7], although the "safety-in-numbers" effect (i.e., an increase in safety with an increase in the number of pedestrians) has also been reported [8].

Various factors influencing pedestrian accidents have been identified: travel time—it is more dangerous to return home [9] or go at night [4,10,11]; place and population, e.g., student campus [12], the elderly [13,14], young pedestrians [15,16], young drivers [17]; visibility of the crossing [18]; perception and listening including mobile phone using [19,20]; the impact of weather [21]; waiting time for the possibility of the cross [22] also at the access

to public transport stops [23,24]; the occurrence of autonomous vehicles [25]; density and differences in the road's environment [26–28].

Sun et al. [7] identify three factors influencing accident rates: location, time, and visibility, whereas Lee et al. [29] listed a number of factors influencing pedestrian behavior, such as: the presence of schools, car ownership, pavement condition, pavement width, bus driving, traffic organization, and the presence of barriers. Solmazer et al. [30] showed significant differences in pedestrian behavior, and other studies show local differences in the perception of pedestrian and driver behavior [31] and the fact that crossing the road also means delays [32]. Ahangar et al. [33] considered the influence of rural drivers' mental patterns for the reasons for the accidents. Herrero-Fernandez et al. [34] studied the influence of the emotional state of pedestrians.

The following research and descriptions have been undertaken: how drivers react to pedestrians and when they reduce speed [3,35]; prediction of threats through models taking into account the speed limit, distance, lighting, visibility, geometry [36]; uncertainties in pedestrian movement [37]; pedestrian-vehicle (driver) interaction based on an advanced pattern-based approach [38]; the impact of MLS data on the assessment of infrastructure safety [39]; similarities and differences on road and rail pedestrian crossings [40]. The perception of the driver and pedestrian as well as other psychological aspects means the occurrence of one of three situations: the vehicle will stop just before crossing, stop earlier, will not stop—there will be a collision [41].

The above observations result in the fact that studies are also undertaken to assess the risk of pedestrian mortality [42]. The speed of pedestrians was also studied [43]. There is a dependence on the probability of a pedestrian's death and the vehicle speed (V). When $V = 30$ km/h, the risk is 5%, when $V = 59$ km/h—50%, and when $V = 80$ km/h—90% [44]. This dependency was previously overestimated [45]. A 1% increase in vehicle speed means an 11% increase in pedestrian mortality [46]. Accordingly, it can be considered that $V = 30$ km/h is a relatively safe speed. Obtaining such a speed value at a pedestrian crossing by a significant group of drivers would allow limiting the consequences of accidents. Such a goal is possible to achieve through the appropriate selection of traffic calming measures, causing the entry to the pedestrian crossing at a speed close to 30 km/h.

In the above description, the limit (critical) values of the instantaneous speed were used. In the analysis of the traffic flow of vehicles, different speed values will appear. Statistical analysis and the use of specific measures such as minimum, maximum and mean value, median, quantiles, skewness, or kurtosis are necessary. For example, the $V_{85}$ quantile (percentile) of speed was used by Wilmots et al. [47]. Some of these measures will be defined in this paper later. However, more commonly used measures (such as mean value) will not be defined.

As "traffic calming measures", we understand a wide range of solutions (activities) aimed at, inter alia, for speed reduction. These can be activities related to the organization of the transport network (e.g., introducing one-way roads, using specific markings). Vehicle speed reduction "devices" are specific solutions in the field of traffic calming measures consisting of introducing building objects (elements) into the road's environment. Examples of such devices are speed tables/bumps, mid-blocks (refuge islands), or raised crossings considered in this paper.

## 2. The Description of Devices Used to Reduce the Vehicles Speed with a Presentation of the Research of Their Effectiveness

Studies on the effectiveness of speed reduction devices are conducted all over the world. Examples of recent works come from: the USA [48], Philippines [49], New Zealand [50], UK [51], Iran [52], China [53,54], Italy [55,56], Turkey [57], and India [58,59]. Some of the studies were carried out in models [57] or in a virtual environment [17,55]. Zeeger and Bushell [60], based on a review and statistics from Europe and the USA, presented examples of devices that improve pedestrian safety. The frequently analyzed ones include: refuge islands [61] or raised crossings [2].

The introduction of new crossing markings has a positive effect on the behavior of drivers [53]. The perception and acceptance of distances between vehicles depend on the length of the pedestrian crossing and its configuration—people who stay on the refugee island accept shorter intervals. The effectiveness of traffic calming devices varies in confrontation with "drivers' psychological risk profiles" [56]. The four solutions for multi-lane roads tested there proved their effectiveness with the exception of the configuration: mid-block plus additional color marking. The number of lanes also affects the behavior of pedestrians. When there are six lanes, there are almost six times more collisions than when there are two lanes and twice as many as when there are four lanes [62]. The refugee islands give a greater reduction of speed when connected with lane removal or narrowings. The single rising of the road has a similar effect as the island [48].

The conducted tests of traffic calming devices show that on raised pedestrian crossings: a road's width has a large impact on speed, mean speed, $\overline{V}$ = 26 km/h, and speed reduction, $\Delta V$ = 43% [63]. In this work, the standard deviation of the speed, $\sigma$ = 7.1 km/h, was identified, and in Aronsson [64], $\sigma$ = 5 km/h. The highest speed reduction applies to speed tables, and then to speed bumps. Barbosa et al. [51] paid attention not to the final value of the speed, but to the range of its reduction. For raised pedestrian crossings, reduction of $V_{85}$ is by 10%, $\Delta V$ is less by 3%, the number of accidents is less by 10% [65], and the median of $V$ is less by 6.5–19.3 km/h [66]. The strongest reduction in speed is with the rising by $H$ = 15 cm [67]. The optimal dimensions of the speed bump were also tested in Turkey using a 1/6 scale model [57]. The study indicated that the optimal dimensions of the speed bump are: 5.0 cm wide and 2.8 cm high. Taking into account the scale of the model, these values should be converted to 30 cm wide and 16.8 cm high.

Alavi [63] presented the following conclusions based on the geometric data of raised pedestrian crossings and on measurements of speed in 96 locations:

- The mean speed is 26 km/h and mean geometric parameters, length, $L$ = 8.96 m and height, $H$ = 10.82 cm,
- The logarithmic function is the most appropriate to describe the relationship between the geometric parameters and the reduction of speed,
- There is a large impact of the road width on speed,
- The following geometric parameters have a significant effect on the speed: ramp gradient, table length, and height.

Basil et al. [68] take into account the mean speed, $\overline{V}$ and the quantile $V_{85}$. For the symmetrical narrowings, no speed reduction was found, and for speed bumps and directional narrowings—a reduction by 21.1 and 19.9 km/h, respectively. The high homogeneity of the reduction was found, standard deviation, $\sigma$ = 10 km/h. The quantile $V_{85}$ is also considered in Ewing [69] and Roess et al. [70]. In these tests, the maximum reduction of speed occurs for the speed bumps (20%) and the minimum reduction is for raised junctions and narrowings.

The results of the Basil et al. research [68] allow estimating the optimal distance between speed reduction devices placed in a series. The optimal distance allows maintaining the recommended speed value along the entire road section. These are the following values:

- 170 m between the speed bumps and 145 m between the speed tables, for a speed of min. 50 km/h;
- 85 m between the speed bumps and 70 m between the speed tables, for a speed of max. 40–45 km/h.

Domenichini et al. [55] showed that it is possible to effectively reduce the speed without discomfort to drivers. Before testing the solutions, it was expected that the design would be able to reduce the speed by 8 km/h. The goal was achieved by introducing frequent devices, spaced on average every 200 m, in the form of raised junctions, raised pedestrian crossings, narrowings, and reduction of the width of traffic lanes. In the case of raised crossings, a speed reduction of 15 km/h was achieved.

In each case, the dominant influence of the geometric parameters of the devices and the manner of their arrangement was found. Geometric parameters determine the value of the speed at which an obstacle is overcome, and the distance between successive devices defines the speed reduction area. For example, for roads with the speed, $V_{85}$ = 40 km/h recorded before the traffic-calming section, on the section equipped with speed reduction devices, the following values were measured: at the site of the deceleration device $V_{85}$ = 27 km/h, 30 m in front of this device $V_{85}$ = 33 km/h, and 30 m behind this device $V_{85}$ = 30 km/h.

Too long speed tables result in a lower reduction of speed [52]. The bigger their height, the lower the speed [58]. The speed tables are characterized by higher speed values than the speed bumps. The devices in the form of single markings fulfill their purpose to a small extent so that the observed speed values are still above the established limit. Similar effects are achieved when the devices are located too far away, i.e., above 225 m [71].

Therefore, the number (density) of devices is important. The spacing of the bumps should be less than 150 m [72]. However, according to Sundo and Diaz [49], drivers accelerate at intervals of more than 100 m. According to Zeeger [73], there are no effects above 200 m, and when the bumps are every 90 m, then $\overline{V}$ = 35.8 km/h along the entire road.

The speed is lower when the ratio of the bump's width ($W_H$) to the road's width ($W_R$) is lower [68]. In conducted measurements, the speed range was from 21.9 km/h to 33.9 km/h, the average speed $\overline{V}$ = 22.3 km/h, a maximum speed equals 51.3 km/h, and minimum speed equals 6.1 km/h.

Falamarzi and Rahmat [52] presented a proposal for the selection of the length of the speed table at a fixed height depending on the assumed speed value. The data from the experiment allowed us to determine the speed (quantile $V_{85}$) and the scope of application for individual lengths of the speed table, i.e., 6.5 m for the speed of 41.5 km/h and 8.5 m for 47.5 km/h. The conducted observations showed that when the table's length was increased by 2 m, the speed increased by about 3 km/h. Due to the comfort of driving and ensuring the safety of people in vehicles, the length of the table should not be less than 2 m for local roads with a speed limit of 40 km/h.

The subject of the research of Gupta [58] was the determination of speed changes at bumps of 75 mm and 100 mm high. The most important research results are as follows:

- 80% of the vehicles slowed down at the hump by 65% compared to the speed on the preceding section,
- The speed at the bump decreases with the increase of its height.

Resuming, a set of geometrical parameters is important to describe the specific devices. Characterizing the most popular (and considered in the next part of the paper) devices, the following parameters are used (see Figure 1): according to the refugee island, *La*—length of refugee island, *Da*—distance between the beginning of the mid-block and pedestrian crossing, *Ha*—width of asylum in refugee island, *S*—slope of narrowing, *Wa*—width of refugee island, *Wl*—width of lane, *W*—width of road; according to the speed table, *Lh*—length of table, *Dh*—distance between table and pedestrian crossing, *Hh*—height of table, *Wh*—width of table, *Wl*—width of lane; according to the raised pedestrian crossing, *Lrc*—length of the raised part of the road, *Rrc*—length of ramp (between raised and non-raised part of road), *Hrc*—height of raising, *Wrc*—width of pedestrian crossing, and *W*—width of road.

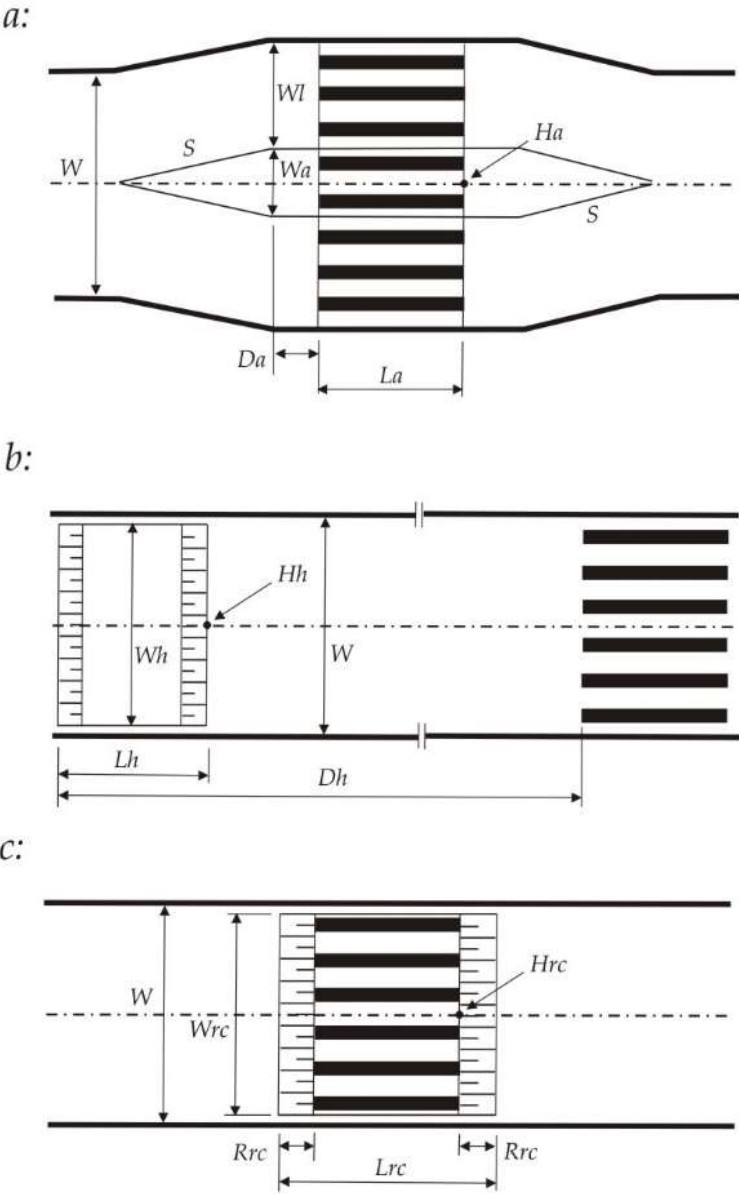

**Figure 1.** Basic parameters of considered devices: (**a**)—refugee island, (**b**)—speed table, (**c**)—raised pedestrian crossing.

### 3. The Studies That Have Been Conducted—Method and Location

This paper presents the assessment of the effectiveness of selected traffic calming devices in terms of reducing the speed of vehicles at pedestrian crossings. Three types of devices are considered: refugee island, speed table, and raised crossing. Measurements were carried out in several locations in Lower Silesia (south-western Poland, Figure 2). There, 105 places were selected, in four areas of different characters of spatial development: rural, small towns, medium-sized towns, and the large city (Wrocław). In each of these places were selected: 5 refugee islands, 10 speed tables, 10–15 raised pedestrian crossings (10 in urbanized areas and 15 in rural areas) while maintaining the necessary size of the measurement sample.

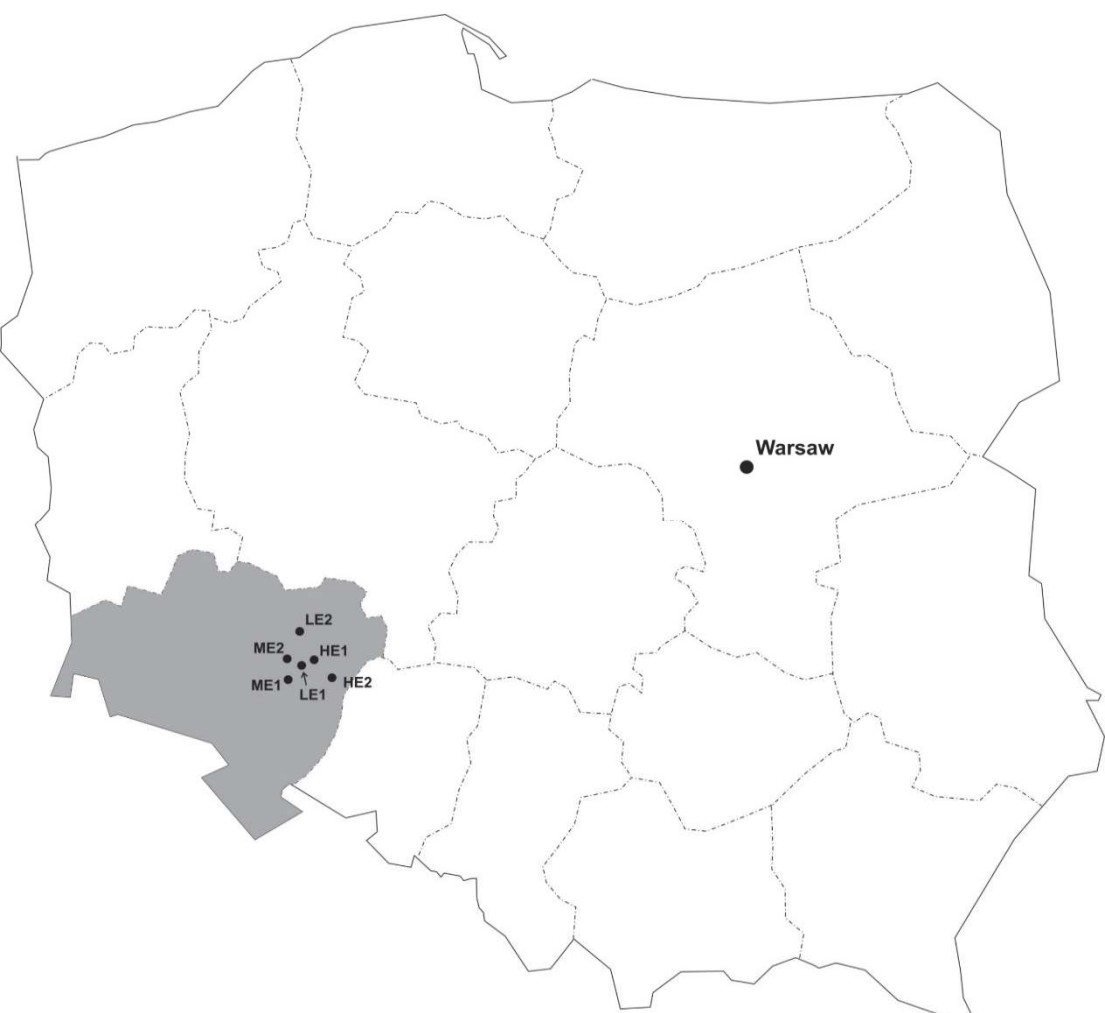

**Figure 2.** Lower Silesia as a part of Poland and location of selected places of measurements.

The first part of the research included the identification of the basic characteristic of traffic (speed and volume) using the SR4 device (Figure 3), which enables the measurement of instantaneous speed, time intervals between vehicles, vehicle length, and, additionally, registration of traffic direction, date and time of measurement. Vehicles are assigned to one of four groups: (1) bicycle, (2) passenger or delivery car, (3) truck or bus, and (4) truck with trailer. The SR4 device is placed in a box mounted directly next to the road, e.g., on a road signpost. The SR4 device collects and processes the data and then sends it electronically to a defined address. The software designed to operate the device provides preliminary statistical processing of the measurement's data, providing information on the values of speed: mean ($\overline{V}$), maximum, quantile $V_{15}$, and quantile $V_{85}$ in all the samples and divided into groups of vehicles. Knowing the value of the quantile $V_{85}$ allows us to identify the problem of moving a significant group of drivers at high speed. The value of quantile $V_{15}$ is used to identify the slowest vehicles that are causing traffic disruption.

The instantaneous speed of the vehicles towards the pedestrian crossing in two cross sections of the road was measured (Figure 4). The first cross section (P0) is located at the access point to the pedestrian crossing, and the second cross section (P1)—at a distance of 60 m from the crossing. This is the stopping distance in front of a hypothetical obstacle. It is determined assuming that 85% of drivers do not exceed the speed of 60 km/h when approaching a pedestrian crossing.

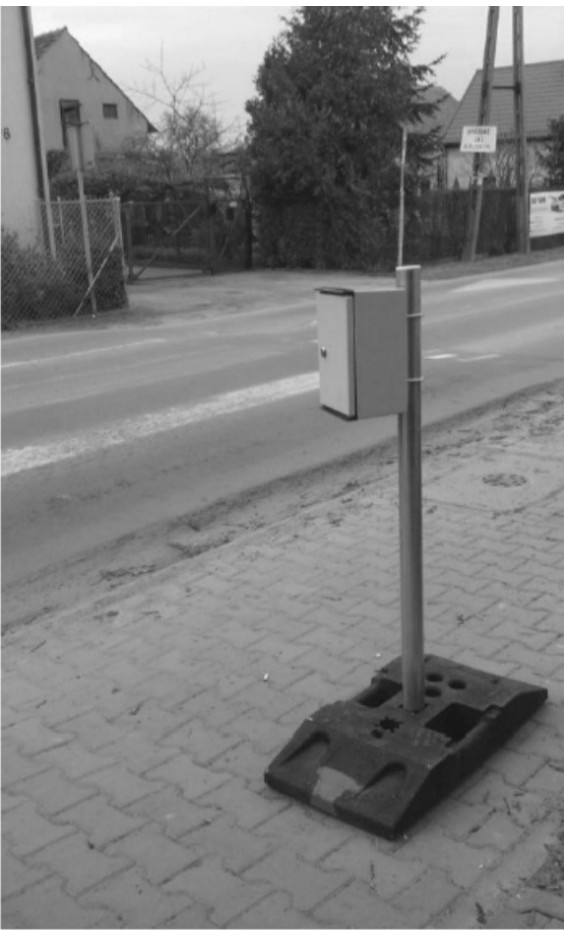

**Figure 3.** SR4 device used in measurements.

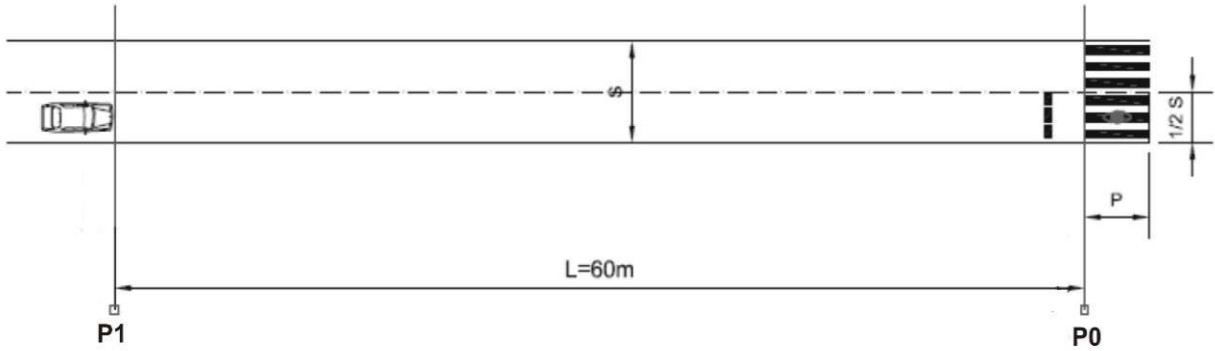

**Figure 4.** The configuration of measurements place.

The speed limits set for urbanized areas apply in most places (in Poland it is 50 km/h); however, there are cases of a local speed limitation of 40 km/h. The speed limits of 30 km/h are introduced in front of the speed bumps and pedestrian crossings. For further analysis, a sample was selected that included 100 passenger cars in each of the designated places with free traffic conditions (the time intervals between vehicles are at least 6 s).

The analyses were performed taking into account: traffic characteristics, constructional details of traffic calming devices, road features, road environment features, and instantaneous speed, in accordance with the measurements taken. This will allow us to start the evaluation of the effectiveness of individual devices (described in the next chapter of the paper).

### 4. The Results of Research and Identification of the Three Categories of Devices

To assess the effectiveness of traffic-calming devices in the context of reducing vehicle speed a several descriptive statistics tools were used. This makes it possible to identify the variability of traffic and the tendency to drive faster or slower than the average. The most important thing is the evaluation of speed differences, the value of which indicates the effectiveness of the selected traffic-calming device. The subject of the analyzes was a comparison of the speed measurement results obtained in two cross sections (P0 and P1) on the places selected for the tests. Here, the results of descriptive analyzes for all the traffic-calming devices selected for testing are presented, without taking into account the division into areas.

The presented summary of measurement's results (Table 1) includes quantile $V_{98}$, quantile $V_{85}$, mean speed, $\overline{V}$ and relative rate of speed's change, $Z_V$ defined according to Formula (1). Additionally are shown: standard deviation, $\sigma$, coefficient of variation, $WZ_V$—Formula (2), skewness, $A$—Formula, (3) and kurtosis, $K$—Formula (4).

$$Z_V = \frac{(V_i^{P1} - V_i^{P0})}{V_i^{P1}} \tag{1}$$

where:

$Z_V$—relative rate of speed's change,
$V_i^{P0}$—mean speed in cross section $P0$ [km/h],
$V_i^{P1}$—mean speed in cross section $P1$ [km/h].

$$WZ_V = \frac{\sigma}{\overline{V}} \tag{2}$$

where:

$WZ_V$—coefficient of variation,
$\overline{V}$—mean speed [km/h],
$\sigma$—standard deviation [km/h].

$$A = \frac{M_3}{\sigma^3} \tag{3}$$

where:

$A$—skewness,
$M_3$—third central moment (the expected value of a third power of the deviation of the random variable from the mean) [km/h].

$$K = \frac{M_4}{\sigma^4} - 3 \tag{4}$$

where:

$K$—kurtosis,
$M_4$—fourth central moment (the expected value of a fourth power of the deviation of the random variable from the mean) [km/h].

A large difference in speed values was identified for individual traffic calming devices. The lowest speed values were recorded on the places with speed tables, the highest on those with refugee islands. The highest reduction of speed values, with high statistical significance, was found for raised pedestrian crossings. This is evidenced by a high positive value of the relative rate of speed's change ($Z_V$). The different values of the coefficient of variation ($WZ_V$) indicate a change in driving style in the neighborhood of a raised crossing, while similar low values for the speed tables indicate a homogeneous driving style and slight speed variation between both cross sections. Similarly, higher $WZ_V$ values indicate the presence of speed modifying factors. Positive values of the skewness ($A$) and kurtosis ($K$) are associated with the tendency to drive at below mean speed and with lower speed dispersion.

**Table 1.** Values of the most important speed characteristics and their changes for the tested traffic calming devices.

| Cross Section | Speed Characteristics | | | | | | | | | |
|---|---|---|---|---|---|---|---|---|---|---|
| | $V_{98}$ [km/h] | $ZV_{98}$ [–] | $V_{85}$ [km/h] | $ZV_{85}$ [–] | $\overline{V}$ [km/h] | $Z\overline{V}$ [km/h] | $\sigma$ [km/h] | $WZ_V$ [–] | $A$ [–] | $K$ [–] |
| | Refugee islands | | | | | | | | | |
| P0 | 65.2 | 0.034 | 56.5 | 0.026 | 48.2 | 0.029 | 8.4 | 0.174 | 0.039 | 0.124 |
| P1 | 67.7 | | 58.3 | | 50.0 | | 8.4 | 0.169 | 0.179 | 0.020 |
| | Speed tables | | | | | | | | | |
| P0 | 44.1 | 0.040 | 36.4 | 0.063 | 30.4 | 0.063 | 6.3 | 0.208 | 0.311 | 0.383 |
| P1 | 46.4 | | 39.5 | | 33.0 | | 6.6 | 0.199 | 0.206 | 0.328 |
| | Raised crossings | | | | | | | | | |
| P0 | 41.7 | 0.250 | 34.7 | 0.228 | 27.7 | 0.286 | 6.6 | 0.238 | 0.314 | 0.245 |
| P1 | 54.7 | | 47.0 | | 39.6 | | 7.3 | 0.185 | 0.214 | 0.381 |

The analyses above make it possible to classify the tested traffic calming devices into one of three categories: high, medium, and low (or no) effectiveness. The classification is based on the range of speed's reduction of quantile 85 ($\Delta V_{85}$). For high-effective devices, $\Delta V_{85} \geq 20$ km/h, for medium-effective devices, $20 \geq \Delta V_{85} \geq 5$ km/h, and for low-effective devices, $\Delta V_{85} \leq 5$ km/h. Two examples of devices from each group are shown further. The range of obtained measurement results is given. Important technical parameters of each device were described. The presented solutions can be found in various areas. The place markings (HE1, HE2, ME1, ME2, LE1, LE2) correspond to the symbols in Figure 2.

The first example of a high-effective device (HE1, Figure 5) is a raised pedestrian crossing located near the school along a county road in a town adjacent to a large city (surroundings of the agglomeration's core). There is a speed limit for the entire area of the town of 40 km/h, and 30 km/h in the vicinity of the pedestrian crossing. The parameters that characterize this traffic-calming device are:

$Lrc$ = 6 m—length of the raised part of the road,
$Rrc$ = 1 m—length of ramp (between raised and non-raised part of road),
$Hrc$ = 0.15 m—height of raising,
$Wrc$ = 6 m—width of pedestrian crossing,
$W$ = 6.5 m—width of road.

Figure 6 shows the speed distribution function for place HE1. For the so-shaped traffic-calming device, the speed reduction was obtained, determined for the quantile $V_{85}$ in the cross section P0 in relation to the cross section P1, $\Delta V_{85}$ = 25 km/h. The cross section P0 has the value of speed very close to the permissible marking (31 vs. 30 km/h), while the cross section P1 has the value of speed much higher than the applicable in the area of the town (56 vs. 40 km/h).

The second high-effective device (HE2, Figure 7) is a combination of refugee island and speed table located along a county road in a small town (approx. 15,000 inhabitants) near the sports facility. There is a speed limit for the town area (50 km/h during the day), and 30 km/h in the vicinity of the pedestrian crossing. The parameters that characterize this traffic-calming device are:

$Lh$ = 3.7 m—length of table,
$Dh$ = 6 m—distance between table and pedestrian crossing,
$Hh$ = 0.08 m—height of table,
$Wh$ = 2 m—width of table,
$Wl$ = 3.5 m—width of lane,
$La$ = 2.5 m—length of refugee island,

*Wa* = 2.5 m—width of refugee island.

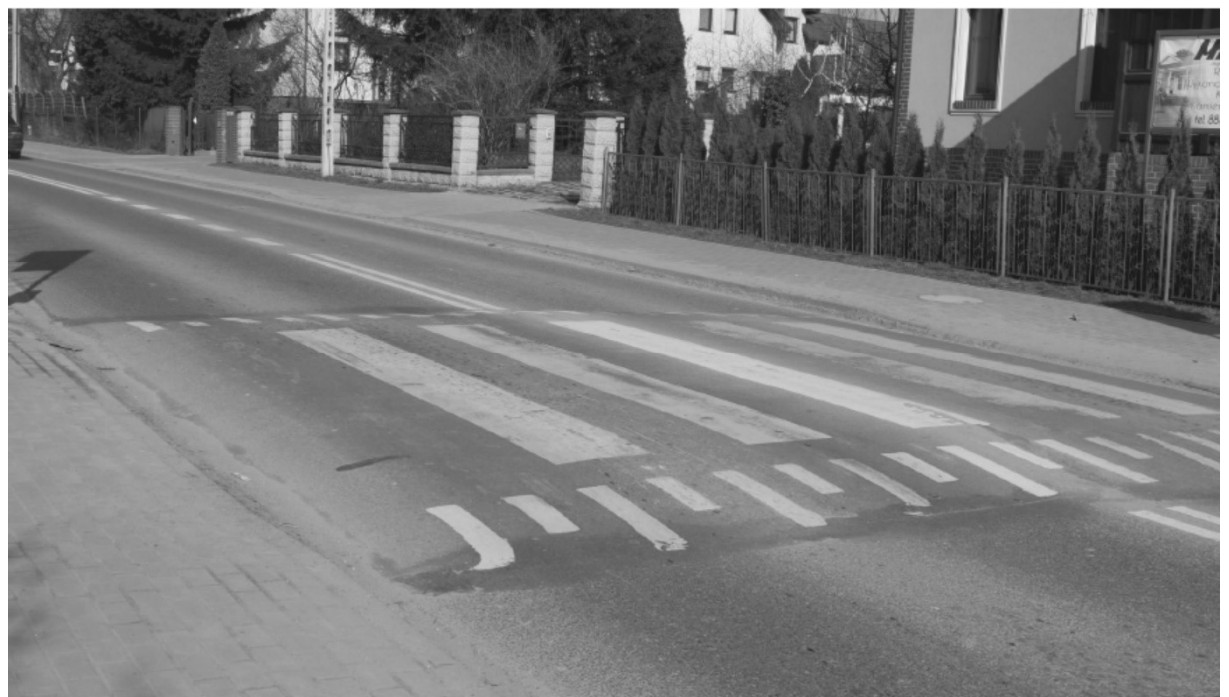

**Figure 5.** The first example of a high-effective device (HE1).

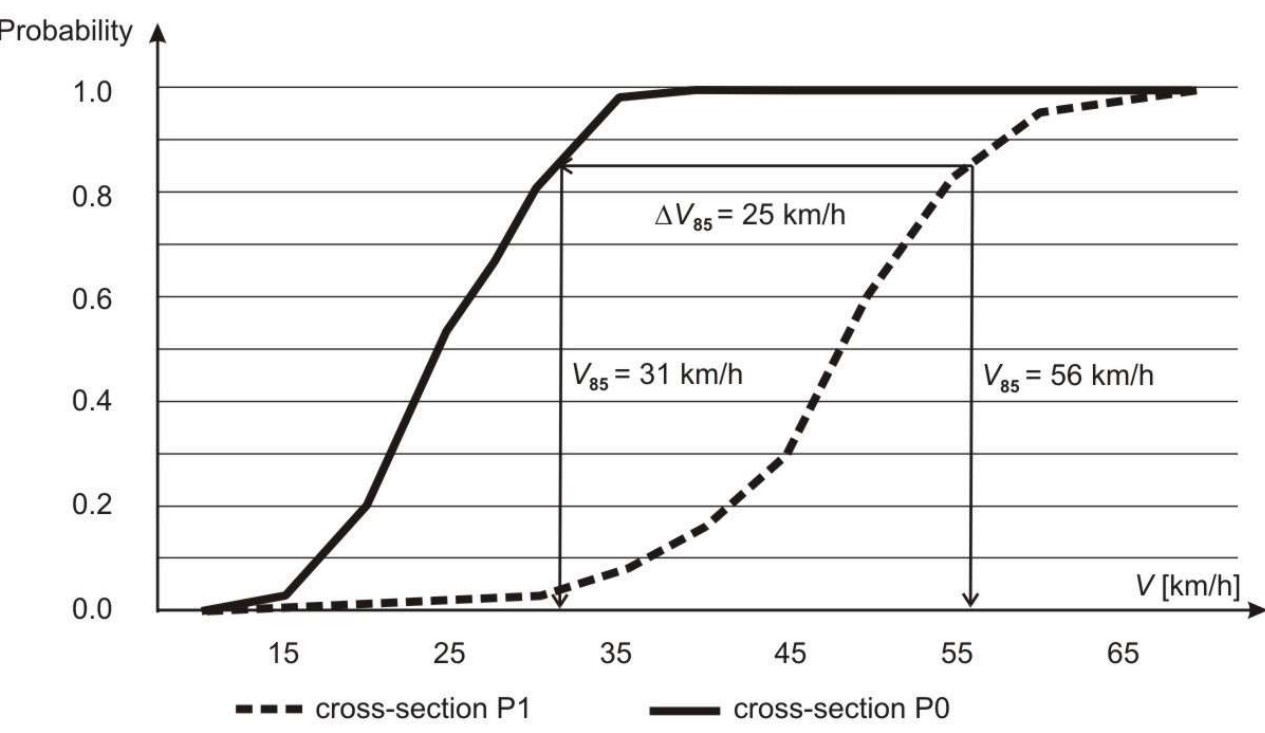

**Figure 6.** The speed distribution function for place HE1.

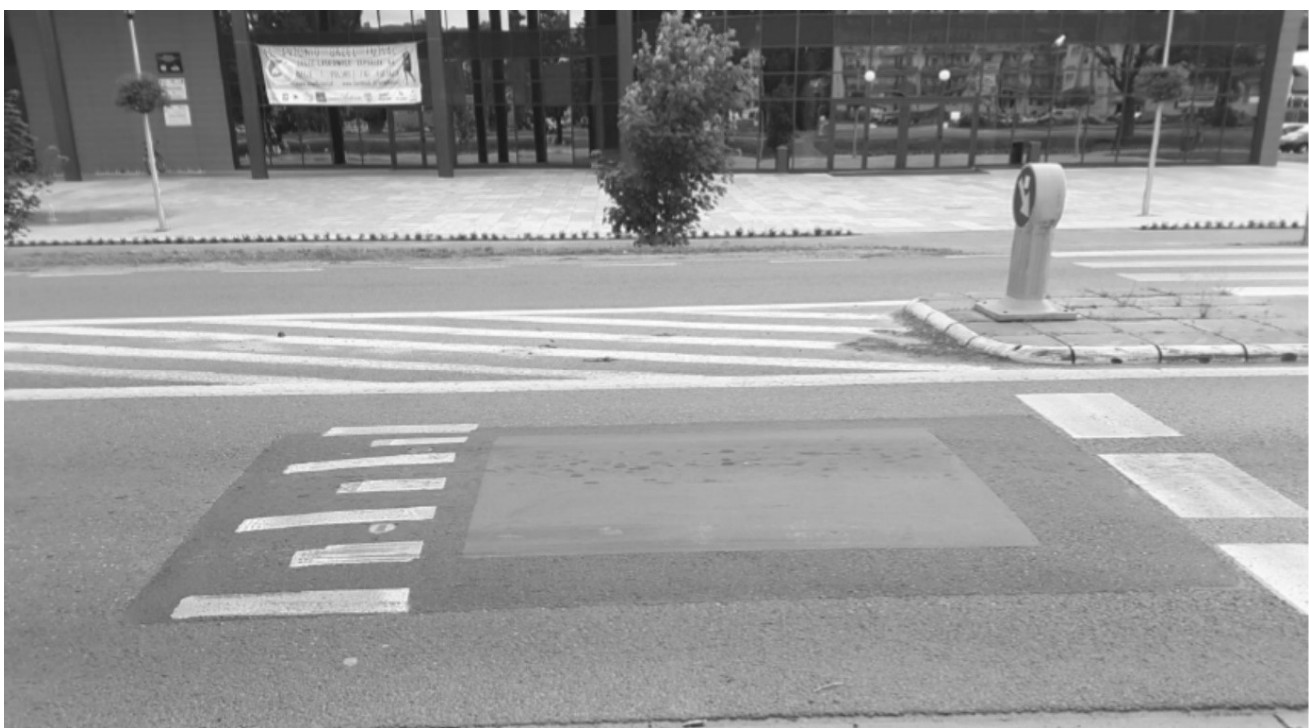

**Figure 7.** The second example of a high-effective device (HE2).

Figure 8 shows the speed distribution function for place HE2. In the case of this traffic-calming device, the speed reduction determined for the quantile $V_{85}$ in the cross section P0 in relation to the cross section P1, $\Delta V_{85} = 21$ km/h was obtained. In cross section P0, the value of speed is slightly higher than the limit (35 vs. 30 km/h), while in cross section P1, the speed value is higher than the applicable (56 vs. 50 km/h), with a dangerous value in the event of an accident.

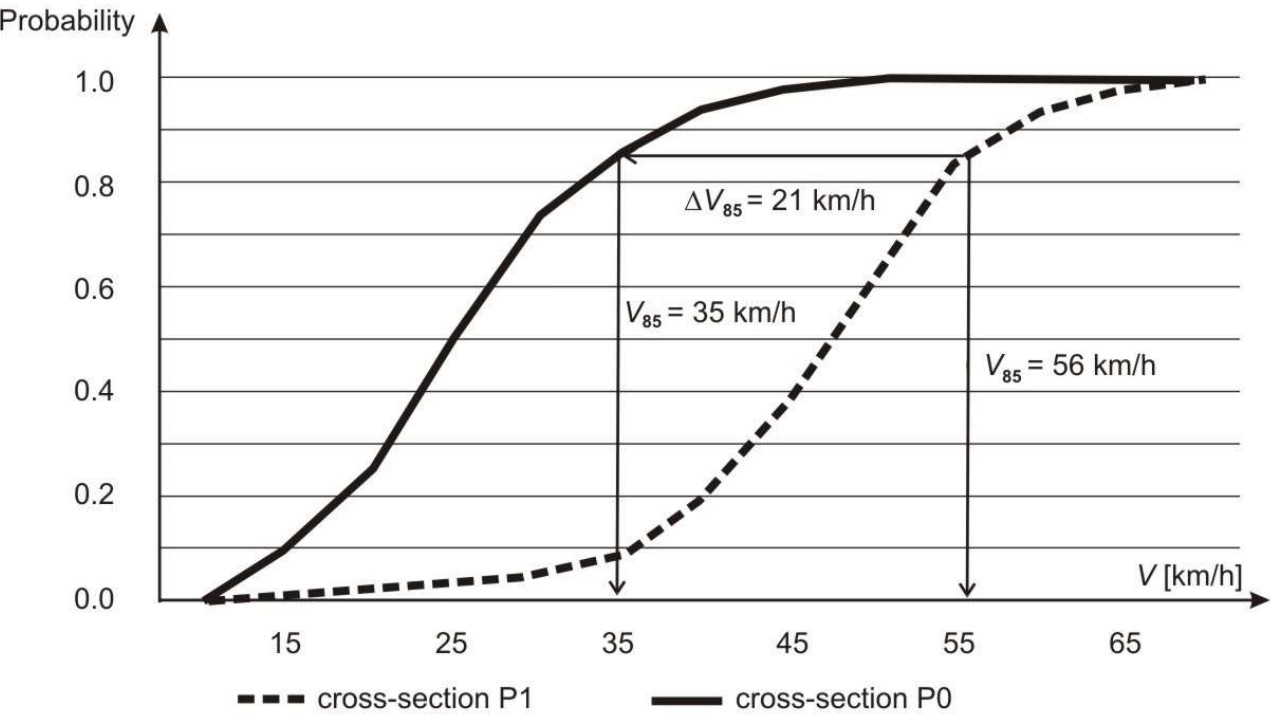

**Figure 8.** The speed distribution function for place HE2.

An example of a site with medium efficiency (ME1, Figure 9) is a raised pedestrian crossing located near the school along a county road in a town adjacent to a large city (surroundings of the agglomeration's core). There is a speed limit for the entire area of the town of 40 km/h and 30 km/h in the vicinity of a raised pedestrian crossing. The parameters that characterize this traffic-calming device are:

$Lrc$ = 7 m—length of the raised part of the road,
$Rrc$ = 1.8 m—length of ramp (between pedestrian crossing and non-raised road),
$Hrc$ = 0.09 m—height of raising,
$Wrc$ = 6.5 m—width of crossing,
$W$ = 7 m—width of road.

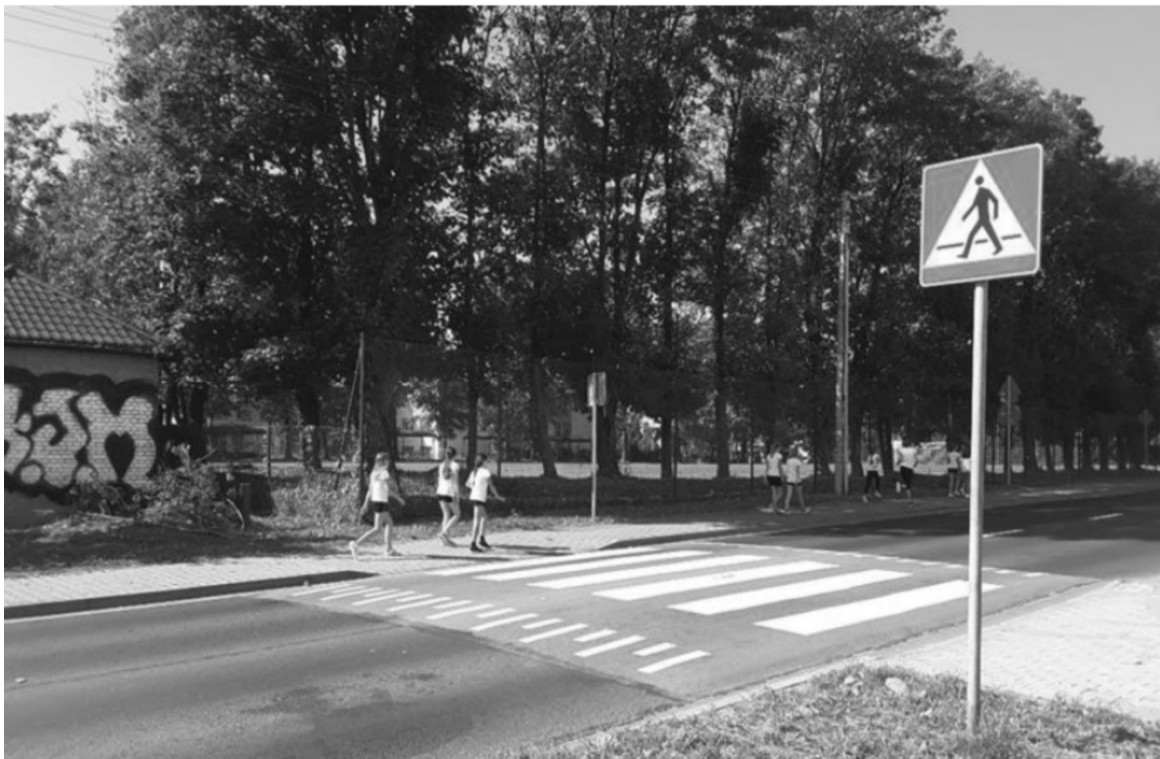

**Figure 9.** The first example of a medium-effective device (ME1).

Figure 10 shows the speed distribution function for place ME1. For such a shaped traffic-calming device, the speed reduction determined for the quantile $V_{85}$ in the cross section P0 in relation to the cross section P1, $\Delta V_{85}$ = 15 km/h. It was found that the value of speed in the cross section P0 exceeds the permissible with the marking by 12 km/h, and the value of speed in the cross section P1 is close to those diagnosed in the place HE1.

The second example of a medium-effective device (ME2, Figure 11) is a series of speed tables located along a municipal road in a large city (approx. 700,000 inhabitants) near the school, at a 30 km/h speed zone. The parameters that characterize these traffic-calming devices are:

$Lh$ = 3.7 m—length of table,
$Dh$ = 25 m—distance between table and pedestrian crossing,
$Hh$ = 0.07 m—height of table,
$Wh$ = 5.5 m—width of table,
$W$ = 6 m—width of road,
$D$ = 100 m—distance between tables.

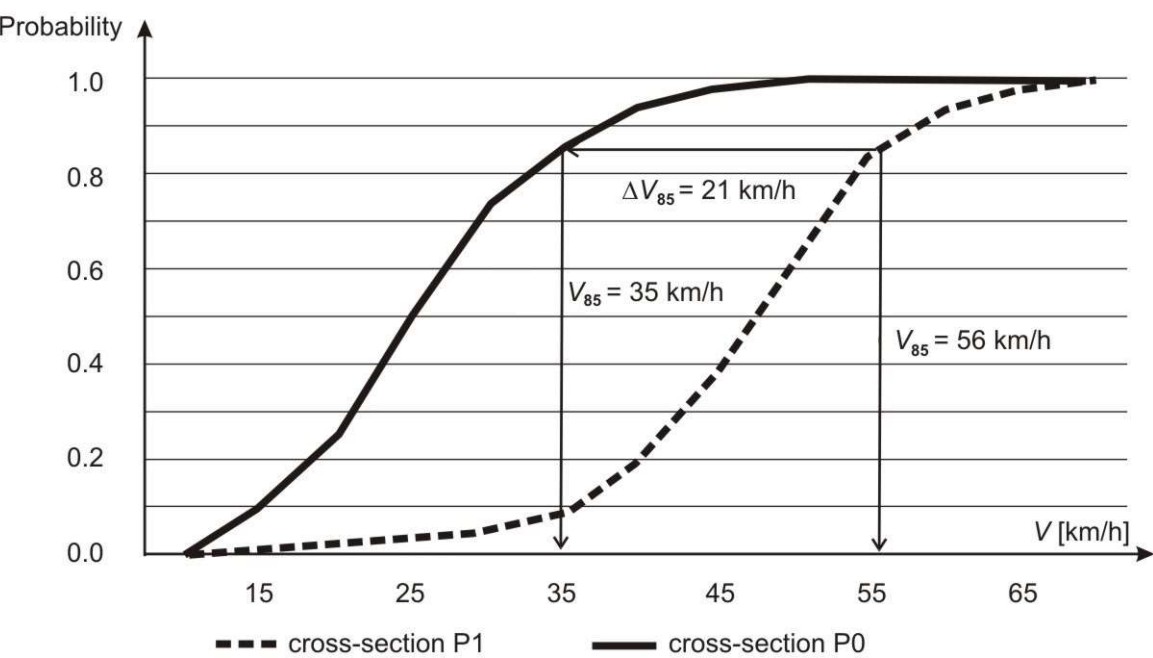

**Figure 10.** The speed distribution function for place ME1.

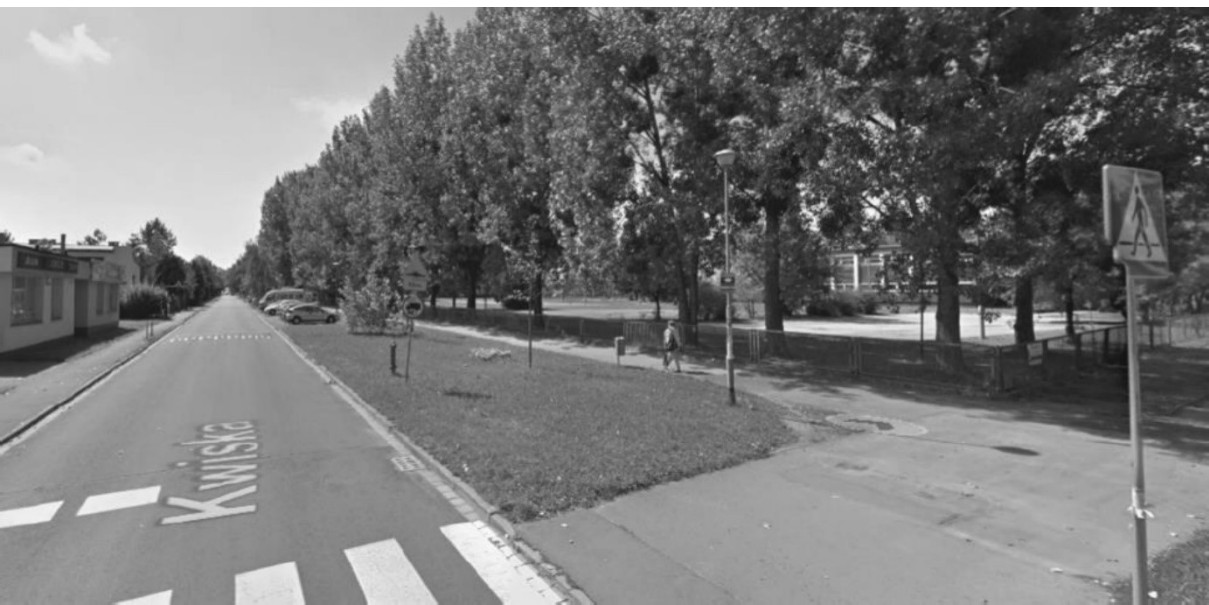

**Figure 11.** The second example of a medium-effective device (ME2).

Figure 12 shows the speed distribution function for place ME2. In the case of the series of speed tables preceding the pedestrian crossing, the speed reduction determined for the quantile $V_{85}$ in the cross section P0 in relation to the cross section P1, $\Delta V_{85} = 10$ km/h. It was found, that the value of speed in the cross section P0 and in the cross section P1 is close to these determined by the markings.

An example of a low-effective device (LE1, Figure 13) is refugee island located along the municipal road separating the housing estate from allotments in a large city (approx. 700,000 inhabitants). The traffic calming device is located adjacent to the bus stop. There is a road-signs speed limit of 40 km/h. The parameters that characterize this traffic-calming device are:

$La = 2$ m—length of refugee island,
$Ha = 0.1$ m—width of asylum in refugee island,

$S$ = 1/10—slope of narrowing,
$Wa$ = 2 m—width of refugee island,
$Wl$ = 3 m—width of lane,
$W$ = 8 m—width of road.

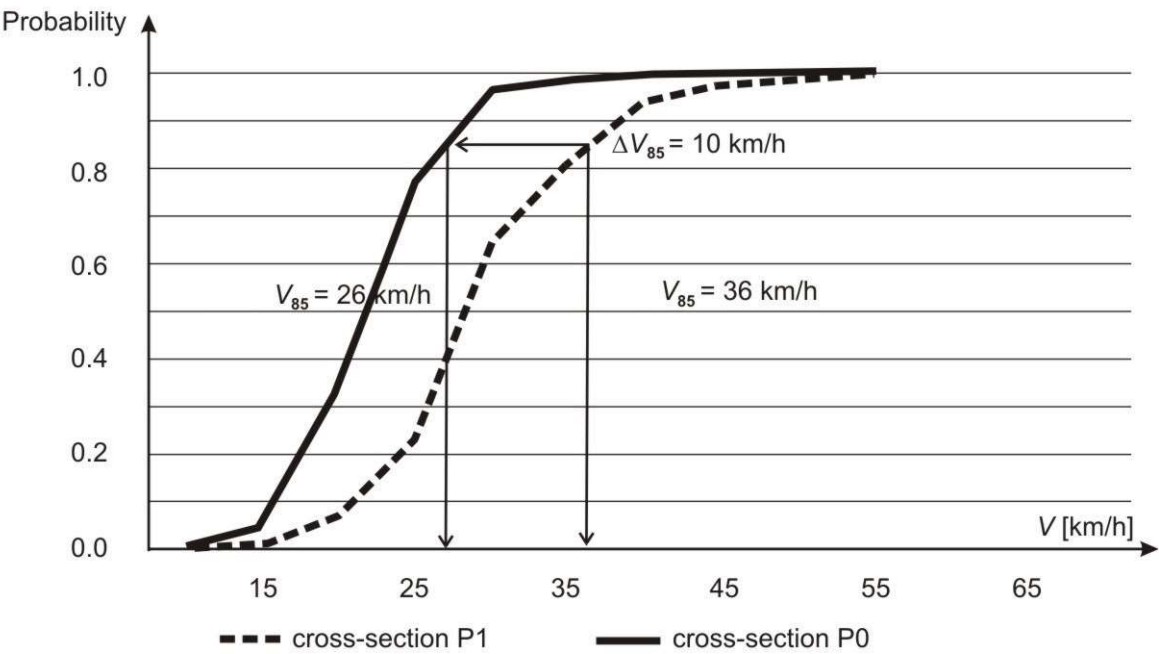

**Figure 12.** The speed distribution function for place ME2.

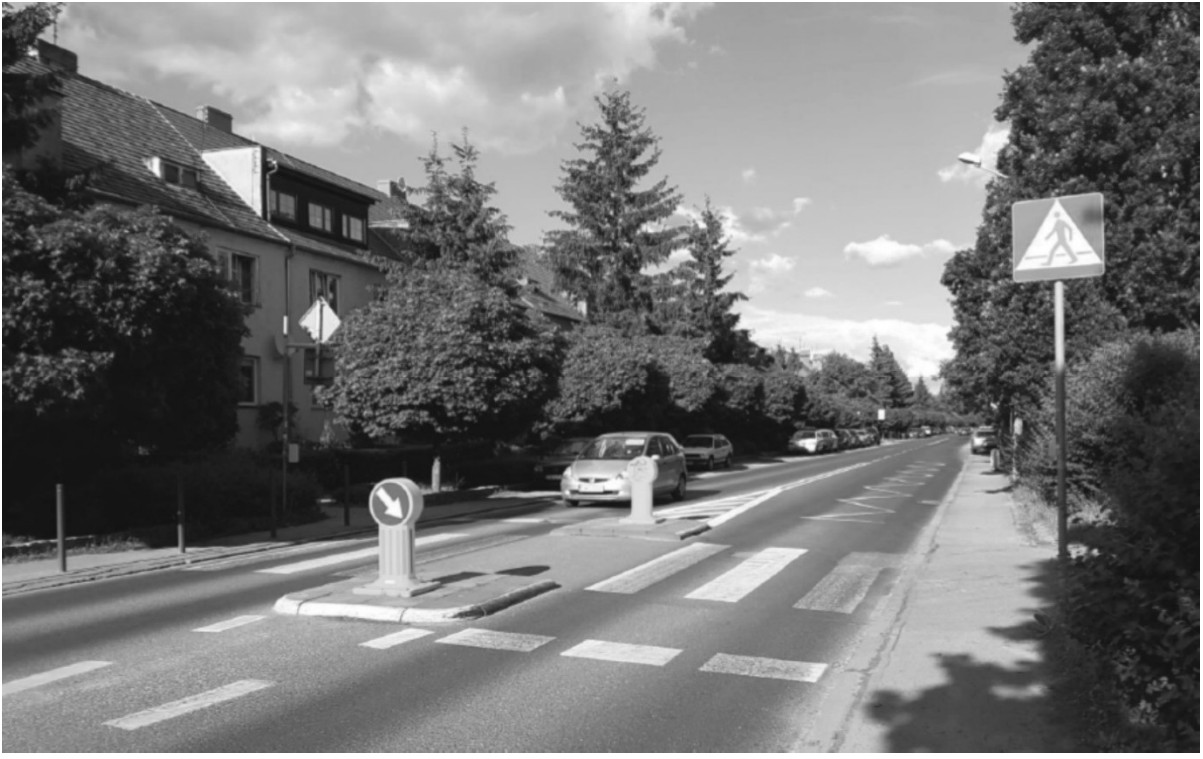

**Figure 13.** The first example of a low-effective device (LE1).

Figure 14 shows the speed distribution function for place LE1. In the case of refuge island, the speed reduction determined for the quantile $V_{85}$ in the cross section P0 in relation to the cross section P1, $\Delta V_{85}$ = 1 km/h was obtained, which is within the measurement's

error limit. It was found that the speeds in both cross sections exceed the value specified with the road signs.

The second example of a low-effective device (LE2, Figure 15) is also refugee island located along the former national road, which is the main transport axis in a small town (approx. 15,000 inhabitants). There is a speed limit of 40 km/h. The pedestrian crossing has additional markings above the road to increase its visibility. The parameters that characterize this traffic-calming device are:

$La$ = 2 m—length of refugee island,
$Ha$ = 0.1 m—width of asylum in refugee island,
$Da$ = 3 m—distance between the beginning of the mid-block and pedestrian crossing,
$S$ = 1/3—slope of narrowing,
$Wa$ = 2 m—width of refugee island,
$Wl$ = 3 m—width of lane,
$W$ = 9 m—width of road.

Figure 16 shows the speed distribution function for place LE2. In the second case of the refugee island, the speed reduction determined for the quantile $V_{85}$ in the cross section P0 in relation to the cross section P1, $\Delta V_{85}$ = 1 km/h was obtained, which is within the measurement error limit. It was found that both speeds' values in both cross sections exceed the limits specified by the road signs.

Conducting research on a selected, relatively homogeneous sample, including three types of devices in 105 precisely selected locations, allowed for the observation and description of general relationships and the introduction of the classification of road safety devices according to their geometrical parameters. The conducted research shows a large variety of geometric and location parameters of the traffic-calming devices used, and thus their effectiveness. Thanks to the adopted research methodology, traffic calming devices were ranked according to their effectiveness in reducing speed. High, medium, and low-effective measures have been identified. The use of low-effective devices is justified solely for purposes other than speed reduction.

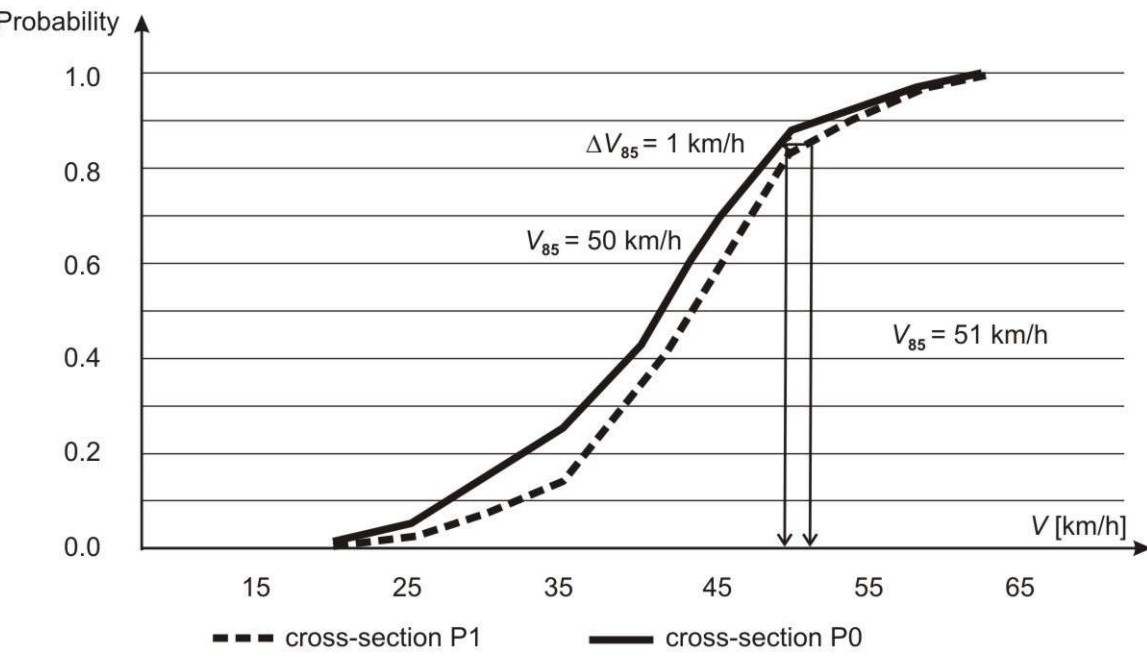

**Figure 14.** The speed distribution function for place LE1.

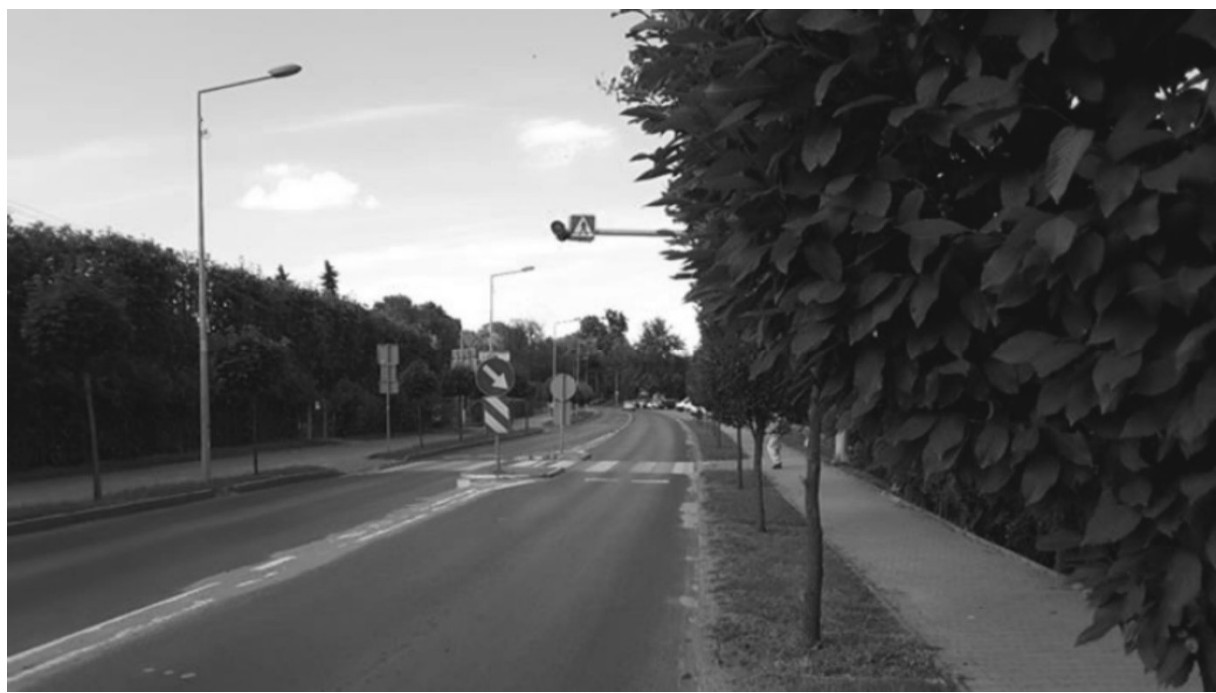

**Figure 15.** The second example of a low-effective device (LE2).

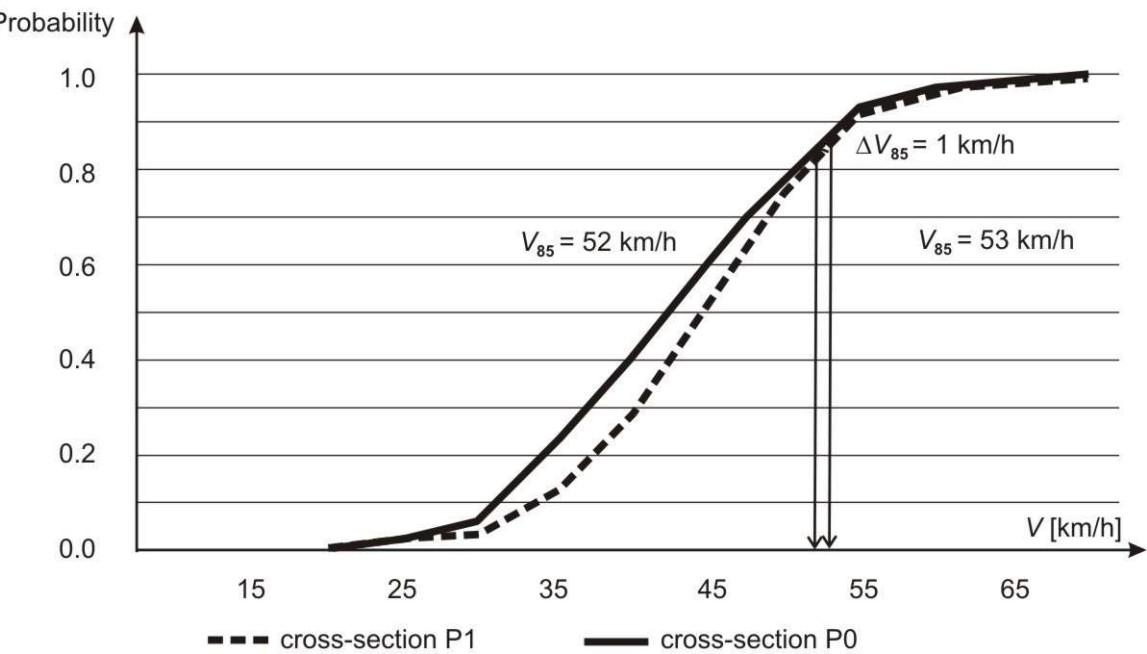

**Figure 16.** The speed distribution function for place LE2.

## 5. Discussion

The research was conducted in order to obtain a homogeneous sample. For this reason, the measurement conditions were standardized in terms of time, weather conditions, and generic structure, which allowed to eliminate the influence of distracting factors on the measurement results. Measurements were carried out during off-peak hours in conditions similar to non-congested traffic. The measurement of geometrical parameters of devices, including height, width, distance, slope, etc., was carried out before starting the traffic measurements.

The research on speed in the vicinity of pedestrian crossings without a traffic light was conducted with specific limitations. The topic covered only road traffic engineering issues without considering the impact on the vehicle and the behavior of pedestrians. The research was conducted in the daytime at pedestrian crossings located outside intersections, on sections of roads prohibited from heavy goods vehicles. The research was carried out in areas with different spatial development features, different cross-sections of the road (with sidewalks, with roadsides). The specific speed parameters, which reflect the driver's reaction to the road, were adopted as the measure of driving behavior. Other aspects could be considered in the next potential researches.

Analyzing the results of speed measurements in 105 locations, no significant differences in speed were observed in areas with different spatial characteristics. Therefore, the classification of devices in each of the areas is the same, which indicates that it is not the spatial features, but the parameters of the device that have a key impact on the behavior of drivers. The conducted research indicates a large variety of geometric and location parameters of the traffic-calming devices used, and thus their effectiveness in reducing speed. Thanks to the adopted research methodology, traffic calming measures were ranked according to their effectiveness in reducing speed. Highly effective and medium-performing ones have been identified.

## 6. Conclusions and Comments

Based on the results of the measurements conducted in all 105 places, the six key conclusions were drawn.

1. There is a wide difference in the impact of the tested devices on speed, and so alters the effectiveness and legitimacy of their use in order to improve pedestrian safety.
2. The greatest reduction of speed in the cross-section P0 is caused by raised pedestrian crossing, as evidenced by positive values of the relative rate of the change of speed $Z_V$.
3. The speed reduction effect along the longer segment of the road (from cross section P1 to cross section P0) is observed when the road is equipped with speed tables.
4. The presence of the refugee island themselves does not significantly reduce the speed of vehicles, the speed value at pedestrian crossings with the island exceeds the (safe) speed limit.
5. Similar values of the coefficient of variation $WZ_v$ in cross section P0 in relation to cross section P1 obtained for speed tables and refugee islands indicate a smaller (than for raised pedestrian crossings) impact of these devices on the change of driving style (including speed) near pedestrian crossings.
6. The majority of drivers reach the value of speed only slightly higher than the safe value (30 km/h) in places equipped with raised pedestrian crossings and speed tables.

The detailed results quoted for selected places show a large difference in the effectiveness of traffic-calming devices depending on their type, location, and geometric parameters. The highest efficiency of raised pedestrian crossings is emphasized even more clearly, both in terms of the speed reduction and its reduction to the values safe for pedestrians in the event of an accident. The effectiveness of devices used in a series and with specific values of geometric parameters (raised parts should be long, but not too long) was also demonstrated.

The above conclusions can be compared with those presented so far in the literature. First of all, the conducted research confirms the high efficiency of raised pedestrian crossings compared to other devices. As in Alavi [63], a speed reduction of 40–45% was achieved with similar geometric parameters. The comparison of the results for the places: HE1 and HE2 confirm the previous observations about the influence of the height of the device on the range of speed's reduction [65,67].

The influence of the density of the devices in a series (and the distance between the device and the pedestrian crossing) was also confirmed. Optimal distances presented in Basil et al. [50], and the observations of Domenichini et al. [55] and Jorgensen et al. [72]

were confirmed by comparing the results obtained in the places: HE2 and ME2 (reduction of speed's values at the level of 40% vs. 25%).

Low effectiveness of the refugee island noted, among others, in [48,55] confirm the results obtained at the places: LH1 and LH2 (almost imperceptible reductions of speed's value). It is also one of the most important conclusions of the research carried out for practical applications: refugee islands are not a safe solution for pedestrians due to the lack of influence on the reduction of vehicle's speed and thus maintaining the risk to pedestrians caused by excessive vehicle speed. The similarity of the observations made for research carried out in different countries allows us to generalize the conclusions and postulate their use in a wider range.

**Author Contributions:** Conceptualization, M.K. and M.M.-P.; methodology, M.K. and M.M.-P.; software, M.K. and M.M.-P.; validation, M.K. and M.M.-P.; formal analysis, M.K. and M.M.-P.; investigation, M.K. and M.M.-P.; resources, M.M.-P.; data curation, M.M.-P.; writing—original draft preparation, M.K. and M.M.-P.; writing—review and editing, M.K.; visualization, M.K.; supervision, M.K.; project administration, M.M.-P. Both authors have read and agreed to the published version of the manuscript.

**Funding:** This research received no external funding.

**Informed Consent Statement:** Not applicable.

**Data Availability Statement:** Not applicable.

**Conflicts of Interest:** The authors declare no conflict of interest.

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
