# Peer review of "The Effectiveness of Selected Devices to Reduce the Speed of Vehicles on Pedestrian Crossings"

_sustainability, doi:10.3390/su13179678_

Round 1
Reviewer 1 Report
There are a few comments thta I have after reading the manuscript.
In the abstract, the authors focus on the necessity of developing speed reduction designs. There is no mention of how these speed devices affect the vehicles specially the tire quality. For example, for a car that has to cross through several pedestrian crossings everyday, the tire's wear and tear is also an important factor.
2. Does the study take into account potential mistakes by pedestrians like not following traffic lights, distraction. If so do these speed reduction devices help or cause more damage?
3. Does the study consider that depending on the shape of the road and other factors different speed control devices have varying efficiency in different areas? In that case, does the high, medium and low classification of the devices change?
4. Some crossings may involve different types of vehicles passing from frequently. Are the effectiveness of the devices still the same or do they change?
5. An assumption was made thta 85% of drivers drive within the speed limit for the Poland study. Is this assumption valid for all times of the day and night?
6. Is speed reduction the only and the best metric for determining the effectiveness of these devices? Other factors like traffic density, road conditions have not been taken into account.
Author Response
Dear Reviewer. Thank you for your opinions and comments. We send our answers in the attached file.

Reviewer 2 Report
- The analysis in Chapter 2 would also be useful to present in graphical form.
- It is recommended to provide a broader summary of Chapter 2.
- It is recommended to expand the section of the methodology, as it is not complete by moving it from section 4, as part of the methodology is presented in section 4.
- It would be useful to provide a broader summary of Chapter 4, as 4 provides important and interesting results but lacks a summary.
Author Response

(The authors gave the same response as above.)

Reviewer 3 Report
This is an ambitious study that measured the effect of devices that promote vehicle speed limits on pedestrian crossings.
It is interesting that the effect of speed reduction is demonstrated by installing various devices.
It's a good paper.
Author Response
Dear Reviewer. Here are no remarks. We very much thank you for such an accommodating opinion.
Round 2
Reviewer 1 Report
I believe the authors have addressed the concerns and comments from the previous reviews. Tge authors mentioned that they have performed these studies during the day and proper weather conditions. I have an additional comment to that. Based on the assumptions how much complexities do the authors feel are needed for this to effectively assist road planners?